# Effects of Microencapsulated Sodium Butyrate, Probiotics and Short Chain Fructooligosaccharides in Patients with Irritable Bowel Syndrome: A Study Protocol of a Randomized Double-Blind Placebo-Controlled Trial

**DOI:** 10.3390/jcm11216587

**Published:** 2022-11-07

**Authors:** Anita Gąsiorowska, Marek Romanowski, Ewa Walecka-Kapica, Aleksandra Kaczka, Cezary Chojnacki, Milena Padysz, Marta Siedlecka, Joanna B. Bierła, Robert E. Steinert, Bożena Cukrowska

**Affiliations:** 1Gastroenterology Department, Medical University of Lodz, ul. Pomorska 251, 92-213 Łódź, Poland; 2Department of Pathomorphology, The Children’s Memorial Health Institute, Aleja Dzieci Polskich 20, 04-730 Warsaw, Poland; 3DSM Nutritional Products Ltd., P.O. Box 2676, 4002 Basel, Switzerland; 4Department of Surgery, Division of Visceral and Transplantation Surgery, University Hospital Zürich, 8091 Zürich, Switzerland

**Keywords:** irritable bowel syndrome, probiotics, prebiotics, butyric acid, study protocol

## Abstract

Irritable bowel syndrome (IBS) is a functional gastrointestinal disease in the pathogenesis of which gut dysbiosis may play an important role. Thus, probiotics, prebiotics, or microbiota metabolites, such as butyric acid, are considered to be effective therapy for IBS. However, there are still no trials presenting the efficacy of these three biotic components administered simultaneously. This study aims to evaluate the effects of the product comprising sodium butyrate, probiotics, and short-chain fructooligosaccharides (scFOS) on the severity of clinical IBS symptoms and the quality of life (IBS-QOL). This is a randomized double-blind placebo-controlled trial conducted in 120 adults with IBS diagnosed according to Rome IV criteria. The intervention group (n = 60) will receive a mixture of the following components: 300 mg of colon-targeted microencapsulated sodium butyrate combined with probiotic *Lactobacillus* strains (*L. rhamnosus* and *L. acidophilus*) and *Bifidobacterium* strains (*B. longum*, *B. bifidum*, *B. lactis*), and 64 mg of prebiotic scFOS. The control group (n = 60) will receive a placebo (maltodextrin). The primary outcomes will be changes in IBS symptoms with the use of the IBS-Severity Scoring System (IBS-SSS), IBS-Global Improvement Scale (IBS-GIS), IBS-Adequate Relief (IBS-AR), and IBS-QOL after 12 weeks of intervention. The secondary outcomes will be the type of stools, patient-recorded symptoms, adverse events, anthropometric and nutritional parameters, and inflammatory cytokine levels. The findings will provide the first evidence of the use of a combination of three biotic compounds in IBS. The study was registered in the clinicaltrials.gov registry under the number NCT05013060.

## 1. Introduction

Irritable bowel syndrome (IBS) is one of the most common functional gastrointestinal disorders characterized by recurrent abdominal pain and altered bowel habits in the absence of an organic disease [1]. The prevalence of IBS ranges from <5% up to 20% depending on the countries or regions and the criteria used [2,3,4]. Depending on the predominant alteration in bowel habits, IBS can be classified into diarrhea-predominant IBS (IBS-D), constipation-predominant IBS (IBS-C), and IBS with mixed bowel habits (IBS-M) [1]. Patients who meet diagnostic criteria for IBS, but whose bowel habits cannot be categorized into one of the three groups should be classified as having unclassified IBS [1].

The pathogenesis of IBS is multifactorial with no single etiology but with a wide spectrum of abnormalities including motility disturbances, visceral sensitivity, altered mucosal immune functions, increased intestinal permeability, disturbances in gut microbiota composition, or altered sensory neuron activation and central nervous system processing [1,5,6,7]. It is currently assumed that gut microbiota dysbiosis may be a potential trigger of IBS inducing most of the pathological conditions described above [6,7,8,9]. Gut microbiota dysbiosis is defined as an imbalance of gut microbial communities and their functional activities within the intestines, which is often associated with disease [10]. Recent research suggests that the gut microbiota is involved in many functions of the human body: it affects nutrient digestion and absorption; it is responsible for immune system function, especially the non-specific immunity provided by the gut-associated immune tissue (GALT), and anti-inflammatory and regulatory activity of GALT; it shapes the intestinal epithelial barrier; affects the microbiota–gut–brain axis [11].

There is evidence of significant alterations in the gut microbiota composition in IBS patients [12,13,14]. Analyses of fecal samples showed decreased proportions of the genera *Bifidobacterium* and *Lactobacillus* and increased ratios of the phylum *Firmicutes*: *Bacteroidetes* in most studies. A recent meta-analysis of microbiome studies found significantly lower abundances of *Lactobacillus*, *Bifidobacterium*, and *Faecalibacterium prausnitzi*, one of the main butyrate producers in the intestine, in IBS patients compared with healthy controls [14]. Moreover, a reduction in butyrate-producing bacteria in patients with IBS-D and IBS-M was found in another study [15]. Butyrate together with acetate and propionate are short-chain fatty acids (SCFAs) with biological effects on the gastrointestinal ecosystem [16]. Butyrate constitutes a key source of energy for enterocytes and has a trophic effect on the colon epithelium, supporting the restoration of the disrupted structural and functional integrity of the epithelial barrier [17]. Butyrate may also affect the composition of intestinal microbiota by inhibiting the proliferation of pathogens and fostering desirable strains [18]. Moreover, butyrate beneficially affects several inflammatory cytokines and myeloperoxidase activity via inhibition of nuclear factor kappa B [19]. These unique properties of butyrate are believed to result in several beneficial effects impacting gut functionality. Given that patients with IBS may suffer from low intestinal butyrate concentrations, as mentioned above, a targeted supplementation to distal parts of the intestine using microencapsulated butyrate has been hypothesized to benefit patients with IBS. In fact, a recent study in IBS patients supplemented with microencapsulated sodium butyrate at a dose of 300 mg per day for 12 weeks found a decrease in the frequency of both abdominal pain and the feeling of incomplete evacuation [20].

Another approach to increase intestinal SCFA concentrations (including butyrate) is the administration of probiotics, prebiotics, or synbiotics (probiotics combined with prebiotics). All of them have the potential to modify intestinal dysbiosis and increase intestinal SCFA production [21], thus bringing therapeutic benefits to IBS patients [22,23,24]. Probiotics are live microorganisms that, when administered at the right dose, have a positive effect on human health [25]. Prebiotics (most often oligosaccharides such as fructooligosaccharides (FOS) or galactooligosaccharides) are substrates that are selectively utilized by host microorganisms conferring a health benefit [26]. There are clinical studies showing the effect of probiotics, prebiotics, and synbiotics in patients with IBS [22,23]. Recently, we reported that a multi-strain synbiotic preparation containing probiotic *Lactobacillus* and *Bifidobacterium* strains and prebiotic short-chain FOS (scFOS) was associated with a significant improvement of IBS-D symptoms and was safe and well-tolerated [27]. These results suggested that the use of this synbiotic offers a benefit for IBS-D patients. However, there are no studies demonstrating the efficacy of a mixture of colon-targeted butyrate and probiotics or synbiotics. Thus, the current study aims to assess the efficacy of colon-targeted microencapsulated sodium butyrate combined with the synbiotic preparation we studied previously in IBS patients.

## 2. Experimental Design

### 2.1. Aim and Hypothesis

The main goal of the study is to assess the effects of the product comprising microencapsulated sodium butyrate and the probiotic mixture of two *Lactobacillus* strains (*L. rhamnosus* and *L. acidophilus*) and three *Bifidobacterium* strains (*B. longum*, *B. bifidum*, and *B. lactis*) and small amounts of prebiotic scFOS on the incidence and severity of clinical symptoms in patients diagnosed with IBS based on the Rome IV criteria. The primary hypothesis is that this mixture supplementation will reduce IBS symptoms and improve the patient’s quality of life.

### 2.2. Study Design

This study will be a prospective, randomized, double-blind controlled trial. One hundred twenty patients from the outpatient clinic at the Gastroenterology Department of the Medical University of Lodz (Poland) will be enrolled in the study. The intervention group (n = 60) will receive the study product containing a mixture of microencapsulated butyric acid, probiotic *Lactobacillus* and *Bifidobacterium* strains, and prebiotic scFOS. The control group will receive a placebo—maltodextrin. The duration of the intervention will be 12 weeks. The study schedule will include five visits: the screening visit, whose purpose is to qualify patients to be enrolled in the study, the baseline visit after up to two weeks after the screening visit, at which participants will be randomized either to the intervention or placebo group, and three follow-up visits (after a 1-week run-in period) at weeks 4, 8, and 12 ± 3 days after the start of the intervention. The visit after eight weeks of intervention will be a phone visit by a researcher, the remaining visits will take place at the clinic. The study protocol diagram is presented in Figure 1.

Once recruited, the participants will receive a diary to fill out on a daily basis. Participants will also be monitored by interviewers via phone every week of the study. The study products will be provided to participants at the baseline visit and at the week 4 visit. At the baseline visit, the participant will be asked not to take the product during the first week (a 7-day run-in period) and to make daily patient-diary entries (containing information about the number and type of stool and the severity of specific IBS symptoms); on the seventh day, a telephone interviewer will instruct each participant to take the study product from the next day onwards.

The patients will be informed of the purposes and conditions of this study and will be asked to sign the relevant informed consent form. The patients will also be informed of the option to refuse and withdraw their consent at any time, without stating the reason or suffering any consequences, and without losing the right to receive further care at the Gastroenterology Department or the outpatient clinic. The patients will receive no compensation for their participation in the study. The study protocol was approved by the Ethics Committee at the Medical University of Lodz, Poland (approval number RNN/213/20/KE). The trial will be conducted in accordance with the ethical principles set out in the Declaration of Helsinki and the guideline on Good Clinical Practice. The study was registered in the clinicaltrials.gov registry (ClinicalTrials.gov identifier: NCT05013060).

### 2.3. Randomization

The subjects will be randomly assigned to study groups (receiving butyrate sodium + probiotics + scFOS or placebo) according to a computer-generated randomization list. In order to ensure a balance in the sample size across groups over time, a block randomization model will be applied, with a block size of 4 [28]. The block size will not be disclosed to the investigators, and both the investigators and the patients will be blinded to group allocation.

### 2.4. Intervention

Study participants will receive a preparation of colon-targeted microencapsulated sodium-butyrate, probiotics, and small amounts of scFOS (Actilight^®^, Beghin Meiji, Marckolsheim, France) or placebo supplied by Biocare Copenhagen, an affiliate of DSM Nutritional Products Ltd. (Table 1).

Microencapsulated sodium butyrate is a colon-release preparation containing a pH-dependent polymer, anionic methacrylate copolymer, E 1207 (EmergoPharm Sp. z o.o. Sp.K., Konstancin-Jeziorna, Poland), which had been recently validated for targeted colon delivery [29]. The dosage forms are hard gelatin capsules (Lonza, Bornem, Belgium), with the placebo capsules containing maltodextrin. All capsules will be identical in size, color, texture, and taste and will be marked as product A or B. The packaging of both products will look identical and will have an inscription containing the title of the study, the approval number of the Ethics Committee, and the expiry date.

## 3. Patients

The study will include 120 adult patients with IBS. IBS will be diagnosed according to Rome IV criteria, i.e., recurrent abdominal pain (felt in the last three months at least one day a week on average), associated with two or more of the following criteria: (1) related to defecation, (2) associated with a change in stool frequency, (3) associated with a change in the form (appearance) of stool [30]. Stool consistency (without the use of any antidiarrheal or laxative agents) will be assessed with the Bristol Stool Form (BSF) scale [31].

### 3.1. Inclusion Criteria

Patients with any of the following forms of IBS will be included:IBS-D—more than 25% of BSF type 6 and 7 stools, and less than 25% of type 1 and 2 stools;IBS-C—more than 25% of BSF type 1 and 2 stools, with less than 25% of type 6 and 7 stools;IBS-M—more than 25% of BSF type 6 and 7 stools and also more than 25% of type 1 and 2 stools.

To be eligible for the trial, subjects must meet all of the inclusion criteria and none of the exclusion criteria, as stated below.

The following inclusion criteria will be used:males and females aged from 18 to 70 years, inclusive;a good physical and mental condition assessed based on the patient’s history and physical examination;laboratory test results (complete blood count, blood chemistry panel) within normal limits or considered not to be clinically significant by the investigator;a voluntarily provided written informed consent;IBS with at least moderate symptom severity, defined as an IBS-Severity Scoring System (IBS-SSS) score of >175 points [32];the ability to strictly adhere to the investigators’ instructions regarding study procedures and protocol requirements.

### 3.2. Exclusion Criteria

Exclusion criteria will include:unclassified IBS;gastrointestinal conditions other than IBS, including clinical or endoscopic diagnosis of gastroenteritis, celiac disease, or inflammatory bowel disease;other diseases, such as respiratory disorders (asthma, chronic obstructive pulmonary disease); cardiovascular disorders, including uncontrolled hypertension (blood pressure > 170/100 mmHg); endocrine disorders, including diabetes mellitus (fasting blood glucose > 11 mmol/L) or thyroid diseases; severe neurological conditions, including psychosis; malignancy; and hepatic or renal impairment;unexplained blood biochemistry abnormalities: serum creatinine levels over twice the upper limit of normal, AST or ALT levels over twice the upper limit of normal;pregnancy or breastfeeding;hypersensitivity to soy or other food allergens;lactose intolerance;a surgical procedure scheduled during the course of the clinical study;the use of gastrointestinal motility stimulants or dietary fiber supplements during the two weeks preceding the clinical study;the use of antithrombotic drugs;current use of gut microbiota-targeted dietary supplements or drugs, such as probiotics, prebiotics, synbiotics, SCFAs, and a refusal to undergo a 1-month washout period;antibiotic therapy during the one month preceding the study;antibiotic use during the study;any drugs, except contraceptive pills or intramuscular contraceptives, hormone replacement therapy (estrogen/progesterone), L-thyroxine, antidepressants at low doses (up to 25 mg of amitriptyline, nortriptyline, or a selective serotonin reuptake inhibitor per day), antihypertensive drugs at low doses (diuretics, angiotensin-converting enzyme inhibitors, angiotensin receptor antagonists) and provided they have been used at a stable dose and for at least one month prior to the study;being included in another clinical study during the previous three months;a history of alcohol or substance abuse;COVID-19 infection, contact with any COVID-19-positive individuals during the previous two weeks.

### 3.3. Withdrawal Criteria

Reasons for the participant to be discontinued from the study by the investigator:informed consent withdrawal,a less than 80% adherence to the study protocol-required product/placebo supplementation,non-attendance at the study visits,the lack of contact with the telephone interviewer,any exclusion criteria found after enrollment,any serious adverse event during the intervention period.

Participants may withdraw from the study at any time without any consequences. In such cases, their study intervention will be stopped and recorded as a “treatment failure”.

Reasons for withdrawal will be recorded by the investigator and saved for data analysis.

## 4. Detailed Procedures

The following measurements will be performed:A physical examination—at each visit;Anthropometric measurements, including weight, height, body mass index (BMI), waist-to-hip ratio (WHR), and arm or calf circumference—at visit 0 and at weeks 4 and 12 of the study intervention;Nutritional status assessment and body composition analysis, including skin fold measurements and bioelectrical impedance analysis (BIA) with the use of a Bodystat machine—at visit 0 and at weeks 4 and 12 of the study intervention;IBS symptom severity assessment, with the use of IBS-SSS [32]—at visit 0 and at weeks 4, 8, and 12 of the study intervention, and with patient-rated (on a Likert scale) symptom severity from patient diaries—weekly;Improvement or worsening of IBS symptoms, with the use of the IBS-SSS, IBS-Global Improvement Scale (IBS-GIS), and IBS—Adequate Relief (IBS-AR) [33]—at weeks 4, 8, and 12 of the study intervention;Adverse events based on the data from telephone interviewers and obtained by the investigators during the whole trial;Quality of life (QOF) assessment, with the use of IBS-QOL questionnaire [34]—at visit 0 and at weeks 4 and 12,Laboratory tests (including complete blood count; liver function tests (ALT, AST); bilirubin, amylase, creatinine, C-reactive protein, glucose, and electrolyte levels at the screening visit; and cytokine (interleukin 6 (IL-6) and macrophage inflammatory protein 1β (MIP-1ß)) levels at visit 0 and at weeks 4 and 12 of the study intervention.

### 4.1. IBS-SSS

The IBS-SSS, developed by Francis, Morris, and Whorwell, is a 5-item survey that focuses on:the severity of abdominal pain (IBS-SSS1),the frequency of abdominal pain over the last 10 days (IBS-SSS2),the severity of abdominal bloating (IBS-SSS3),dissatisfaction with bowel habits (IBS-SSS4),interference with quality of life over the past 10 days (IBS-SSS5) [32].

Subjects will respond to each question on a 100-point visual analog scale. Each of the five questions generates a maximum score of 100 points, and the total scores range from 0 to 500, with higher scores indicating more severe symptoms.

### 4.2. IBS-GIS

The IBS-GIS is a tool for assessing patient-rated changes in IBS symptom severity using a 7-point Likert scale, ranging from symptoms substantially worse (1 point) to substantially improved (7 points) [33]. Participants will be asked to answer the question “Have you felt any change in the severity of your symptoms over the past 7 days compared to how you felt before the medicine was taken?” The answers will be recorded based on a 7-point scale:1 point—“I feel that the symptoms have worsened significantly”;2 points—“I feel that the symptoms have moderately worsened”;3 points—“I feel that the symptoms have slightly worsened”;4 points—“I feel no change”;5 points—“I feel a slight improvement”;6 points—“I feel moderate improvement”;7 points—“I feel significant improvement”.

IBS-GIS scores of >4 indicate an improvement, scores of <4 indicate a worsening, and a score of 4 indicates no change in symptom severity.

### 4.3. IBS-AR

IBS-AR is a dichotomous single-item questionnaire that asks participants “Over the past week, have you had adequate relief of your IBS symptoms?” The answer can be YES or NO.

### 4.4. BSF Scale

The type of stools will be assessed using the BSF Scale that classifies feces into seven types: types 1–2 indicate constipation, types 3–4 are “normal” stools, and types 5–7 indicate diarrhea [31].

### 4.5. IBS-QOL

IBS-QOL is a 34-item measure assessing the degree to which IBS interferes with patient quality of life [34]. Each item is rated on a 5-point Likert scale, thus, yielding a total score theoretically ranging from 34 to 170 points, with higher scores indicating worse QOL.

### 4.6. Patient’s Diary

Participants will be asked to fill out a diary every day. The diary will contain information on the number of bowel movements per day, the type of stools, and the severity of the following symptoms: abdominal pain, bloating, fecal urgency, and feeling of incomplete evacuation. The type of stools will be assessed using the BFS scale. Specific IBS symptoms, except for the feeling of incomplete evacuation, will be assessed using a patient-rated 5-point Likert scale, with 0 points indicating no symptoms, and 1–4 points indicating various levels of symptom severity, with higher scores indicating worse symptoms. The feeling of incomplete evacuation will be assessed using a 2-point scale, with 0 points indicating no such feeling and 1 point indicating the feeling of incomplete evacuation.

### 4.7. Telephone Interview

Telephone interviewers will call participants once a week to monitor study preparation intake and daily diary entries, and to collect data on side effects and the use of other medications, particularly antibiotics. Adverse events will be evaluated on a scale from 0 (no) to 1 (yes). Participants who take antibiotics will be withdrawn from the study.

### 4.8. Study Endpoint Definition

The prospective defined primary efficacy outcome will be changes in IBS symptoms and quality of life measured with the use of the IBS-SSS, IBS-GIS, IBS-AR, and IBS-QOL questionnaires.

Secondary efficacy variables will include:the number and type of stools assessed with the BSF scale,pain, bloating/abdominal distension, stool urgency, feeling of incomplete evacuation after a bowel movement assessed with a patient-rated Likert scale,adverse events,anthropometric measurements and BMI,body composition,cytokine (IL-6, MIP-1ß) levels.

### 4.9. Adverse Events

This is a minimal-risk study that involves the use of a commercially available dietary supplement. Thus, we do not anticipate any adverse events related to the study protocol. Nonetheless, the subjects will be systematically monitored for adverse events in person and via telephone interviews. They will be instructed to contact investigators if any side effects occur. All adverse events will be reported to the medical director of the Medical University Hospital in Lodz (Poland) within 24 h.

### 4.10. Statistical Analyses

#### 4.10.1. Sample Size Calculation

Based on the effects observed in the previous study, in which we used a similar probiotic and prebiotic combination, but without microencapsulated sodium butyrate, in IBS patients [27], we have estimated that a sample size of 35 participants per group would be required to reach clinical significance. With an estimated drop-out ratio of about 20%, the total sample size would need to be 80 participants, which would ensure 80% power to demonstrate significance with a *p*-value < 0.05. To further enhance the statistical power, the sample size was ultimately set to 60 participants/cases per group, with a total of 120 patients to be included in the study.

#### 4.10.2. Statistics

The intergroup and intragroup differences in age, anthropometric parameters, and IBS symptom data across subsequent visits or weekly data from patient diaries or telephone interviews will be evaluated with two-sided paired or unpaired *t*-tests or repeated measures (RM) ANOVA after checking for equality of variances and normality of distribution using a Shapiro–Wilk test. If the data is not normally distributed, two-sample Wilcoxon paired or unpaired signed-rank tests will be used. Fisher’s exact test will be used for analyses of variables presented as percentages. The threshold of significance for all analyses will be set at α = 0.05.

## 5. Expected Results and Discussion

To the best of our current knowledge, this is the first randomized, double-blind, and placebo-controlled trial assessing the role of a mixture of colon-targeted microencapsulated sodium butyrate, probiotics, and small amounts of prebiotics in IBS patients. We believe that such a combination will help restore the disturbed functionality of the intestinal microbiota and increase intestinal levels of butyrate, which may benefit patients with IBS. The beneficial effects of probiotics and synbiotics in IBS patients were demonstrated in several randomized clinical trials [22,23,24,35], but only one study showed the superiority of butyric acid supplementation in comparison with placebo [20]. The efficacy of biotics in IBS depends on preparation composition, dose, and duration of administration [36]. In the present study, we plan to use a combination of sodium butyrate with a probiotic mixture and prebiotic scFOS. We have decided on this composition taking into account our previous research on the effects of a synbiotic product with the same composition of probiotic strains and prebiotic scFOS. In that previous randomized placebo-controlled clinical trial we demonstrated that this specific probiotic mixture containing three *Bifidobacterium* species (*Bifidobacterium lactis*, *Bifidobacterium longum*, *Bifidobacterium bifidum*), two *Lactobacillus* species (*Lactobacillus rhamnosus*, *Lactobacillus acidophilus*), and scFOS significantly decreased the total IBS-SSS score in IBS-D patients [27]. This beneficial effect was mainly achieved by improvement in bloating; however, no effect on the perception of pain was observed. Given that butyric acid has been shown to decrease the severity of both abdominal pain and the feeling of incomplete evacuation [20], we hypothesize that the addition of colon-targeted microencapsulated sodium butyrate plus small amounts of scFOS to this specific probiotic mixture would further enhance the effectiveness of the intervention against a wider range of clinical symptoms.

Changes in gut microbiota composition differ depending on the type of IBS [14]. In a study by Pozuelo et al., a lower abundance of butyric acid-producing bacteria was mainly observed in patients with IBS-D and IBS-M [15]. In contrast, El-Salhy et al. [37] found propionate and butyrate levels to be reduced in IBS-C patients, and butyrate levels to be increased in IBS-D patients in comparison with the control group. We plan to enroll patients with all three types of IBS (IBS-D, IBS-C, and IBS-M). Thus, the current study will allow us to determine in which type of IBS patients the study combination of butyrate sodium, probiotics, and small amounts of scFOS has the best effectiveness.

In our study, we have decided to administer the butyrate sodium + probiotics + scFOS preparation for 12 weeks, which allows us to expect an improvement in specific IBS symptoms measured with the use of international scales (IBS-SSS, IBS-GIS, IBS-AR) and patients’ QOL. As SCFAs (including butyric acid) have been estimated to provide about 10% of the total dietary energy supply in humans [38] and may also stimulate lipogenesis [39] we have planned to assess the impact of SCFA supplementation on body composition, anthropometric measurements, and BMI. We also suspect that a combination of sodium butyrate, probiotics, and small amounts of prebiotics will affect the inflammatory processes present in IBS patients, which we will be able to demonstrate by analyzing selected serological biomarkers (IL-6, MIP-1ß).

Our study has many strengths, most notably the design of a randomized, double-blind placebo-controlled trial, weekly monitoring of participants by interviewers, and assessment of primary endpoints on international scales. However, there are also some limitations. Due to budget constraints, we do not plan to assess the changes in microbiome composition, which may allow us to better explain the mechanism of clinical effects as well as the post-interventional visit to assess the longevity of the response to the intervention. Additionally, the amount of prebiotic scFOS that will be used in this trial is much lower than the amounts used in previous trials, and these small amounts may not be sufficient to have a meaningful physiological effect. This limitation was dictated by the practical size of the capsules. The inclusion of more scFOS would result in too large a capsule size. Nonetheless, we expect the inclusion of colon-delivered microencapsulated sodium butyrate to compensate for the small amounts of scFOS and possibly even show additional beneficial effects. It is important to emphasize that butyric acid is the topic of ongoing discussions as an SCFA and a beneficial metabolite of gut microbiota potentially useful as a postbiotic, although there is currently no consensus [40,41].

## Figures and Tables

**Figure 1 jcm-11-06587-f001:**
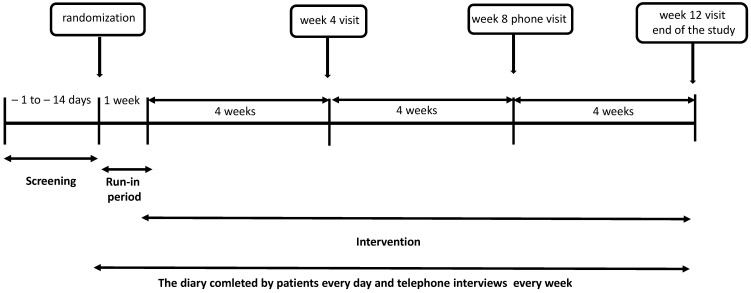
The study protocol diagram.

**Table 1 jcm-11-06587-t001:** Composition and doses of the individual components of the product administered to patients.

Active	Dose per Capsule *
Microencapsulated sodium butyrate	300 mg (equal to 150 mg sodium butyrate)
*Bifidobacterium lactis* DSMZ 32269	5.2 × 10^8^ CFU
*Bifidobacterium longum* DSMZ 32946	1.0 × 10^8^ CFU
*Bifidobacterium bifidum* DSMZ 32403	1.0 × 10^8^ CFU
*Lactobacillus acidophilus* DSMZ 32418	1.4 × 10^8^ CFU
*Lactobacillus rhamnosus* FloraActive19070-2	1.4 × 10^8^ CFU
scFOS	64 mg

* Patients will receive two capsules per day, which will be consumed orally 30 min after a meal, one capsule after breakfast, and one after dinner, for a period of 12 weeks. CFU = colony forming units.

## Data Availability

The data presented in this study will be available on request from the corresponding author. The data will not be publicly available due to privacy protection issues.

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
