# Peer review of "Effects of Microencapsulated Sodium Butyrate, Probiotics and Short Chain Fructooligosaccharides in Patients with Irritable Bowel Syndrome: A Study Protocol of a Randomized Double-Blind Placebo-Controlled Trial"

_jcm, 2022, doi:10.3390/jcm11216587_

Round 1
Reviewer 1 Report
1. Please revise the authorship
2. English language revision is required, for ex:
- line 23 - the word "formulation" or line 27 - the word "preparation" maybe there are not the most suitable words
- line 25 - 'is a randomized...'
- line 26 - 'according to....'
- line 26 - 30 - please restate
- the study design is not very clear formulated, please restate in order to be more comprehensive
- line 304, 305 - please restate
Author Response
Authors would like to thank you for review of our manuscript and all comments and suggestions which allow us to improve the paper. All changes made in the manuscript are marked in red. Below are our answers to comments:
- Please revise the authorship
Thank you for this suggestion. The authorship was corrected (line 463-465)
- English language revision is required
English language was corrected by the professional English translator, and changes suggested by the Reviewer were implemented:
- line 23 - the word "formulation” and line 27 - the word "preparation" were changed into word “product” or restated
- line 25 –has been changed to 'is a randomized...'
- line 26 – has been changed to 'according to....'
- line 26 - 30 – has been restated. Now it is: “The intervention group (n=60) will receive a mixture of the following components: 300 mg of colon-targeted microencapsulated sodium butyrate combined with probiotic Lactobacillus strains (L. rhamnosus and L. acidophilus) and Bifidobacterium strains (B. longum, B. bifidum, B. lactis), and 64 mg of prebiotic scFOS. The control group (n=60) will receive placebo (maltodextrin).”
- the study design was restated to be more comprehensive. Now it is (lines 113-124): “This study will be a prospective, randomized, double-blind controlled trial. One hundred twenty patients from the outpatient clinic at the Gastroenterology Department of the Medical University of Lodz (Poland) will be enrolled in the study. The intervention group (n = 60) will receive the study product containing a mixture of microencapsulated butyric acid, probiotic Lactobacillus and Bifidobacterium strains, and prebiotic scFOS. The control group will receive a placebo - maltodextrin. The duration of the intervention will be 12 weeks. The study schedule will include five visits: the screening visit, whose purpose is to qualify patients to be enrolled in the study, the baseline visit after up to 2 weeks after the screening visit, at which participants will be randomized either to the intervention or placebo group, and three follow-up visits (after a 1-week run-in period) at weeks 4, 8, and 12±3 days after the start of the intervention. The visit after 8 weeks of intervention will be a phone visit by a researcher, the remaining visits will take place at the clinic.”
- line 304, 305 were restated. Now it is (line 313-315): “The IBS-GIS is a tool for assessing patient-rated changes in IBS symptom severity using a 7-point Likert scale, ranging from symptoms substantially worse (1 point) to substantially improved (7 points) [33].”
Reviewer 2 Report
1. Please explain why microencapsulated sodium butyrate, probiotics and short chain reconstoligosaccharides are made into mixture supplementation? Do you have the basis for preliminary research?
2. How many subjects are expected to be included?
3. What are the potential adverse events? How to deal with it?
Author Response
Authors would like to thank you for review of our manuscript and all comments and suggestions which allow us to improve the paper. All changes made in the manuscript are marked in red. Below are our answers to all comments:
- Please explain why microencapsulated sodium butyrate, probiotics and short chain oligosaccharides are made into mixture supplementation? Do you have the basis for preliminary research?
In the revised version of the manuscript we added the information why we decided to use the mixture of specific probiotic strains+scFOS +butyric sodium in the section “Expected Results and Discussion” (lines 415-428).
We explained that “We have decided on this composition taking into account our previous research on the effects of a synbiotic product with the same composition of probiotic strains and prebiotic scFOS. In that previous randomized placebo-controlled clinical trial we demonstrated that this specific probiotic mixture containing three Bifidobacterium species (Bifidobacterium lactis, Bifidobacterium longum, Bifidobacterium bifidum), two Lactoba-cillus species (Lactobacillus rhamnosus, Lactobacillus acidophilus), and scFOS significantly decreased the total IBS-SSS score in IBS-D patients [27]. This beneficial effect was mainly achieved by improvement in bloating; however, no effect on the perception of pain was observed. Given that butyric acid has been shown to decrease the severity of both abdominal pain and feeling of incomplete evacuation [20] we hypothesize that an addition of colon-targeted microencapsulated sodium butyrate plus small amounts of scFOS to this specific probiotic mixture would further enhance the effectiveness of the intervention against a wider range of clinical symptoms”
- How many subjects are expected to be included?
We plan to include 120 subjects, and this information was given in section “ Sample size calculation” (line 394). The question about the number of study participants made us realize that such information is not sufficient and in the revised version of the manuscript, we added the planned number of study participants to the abstract (line 26), and the sections “Patients” (line 193) and "Study design" (lines 113-114).
- What are the potential adverse events? How to deal with it?
Thank you for this question. In the revised version of the manuscript we added the section “Adverse events” (lines 377-383), which is as follows:
“3.9 Adverse events
This is a minimal risk study that involves the use of a commercially available dietary supplement. Thus, we do not anticipate any adverse events related to the study protocol. Nonetheless, the subjects will be systematically monitored for adverse events in person and via telephone interviews. They will be instructed to contact investigators if any side effects occur. All adverse events will be reported to the medical director of the Medical University Hospital in Lodz (Poland) within 24 hours.”
Reviewer 3 Report
Irritable bowel syndrome (IBS) is a major health issue and the cause of a large proportion of outpatient presentations in a general gastroenterology clinic. To date, no convincing therapy exists and care for these patients remains a challenge. Therefore, the study protocol proposed by Gasiorowska A and colleagues is definitely adressing an urgent issue.
The authors propose to study the effects of a synbiotic-postbiotic intervention in IBS patients for the duration of 12 weeks. The randomised placebo-controlled trial design is definitely a benefit in a field that is full of observational studies. The authors also include a run-in period to establish baseline measurements. However, a postinterventional observational period (e.g. 4 -8 weeks after the intervention) would potentially provide insights into durability of response to the combined intervention.
The trial plan in itself is clear and detailed and the statistical considerations presented are sound. A sample size and power calculation has been performed.
Although the authors state, that microbiome analysis is beyond the budgetary constraints of the present project, it might be reasonable to still sample feces from the trial participants at least at baseline. This could be valuable aterial for future analyses. Analysis of feces could also include other aspects beside microbiome composition, e.g. SCFA content. A potential workaround for the costs associated with next-generation metagenomic sequencing would be to conduct targeted RT-QPCR experiments for specific bacteria known to be affected by the synbiotic/postbiotic intervention.
In summary, while I feel that no changes to the protocol are required for this study to provide interesting insights, I would invite the authors to consider the following potential amendments to their trial:
1. Include a post-interventional visit to assess longevity of response to the intervention
2. Include baseline (and possibly also week12 and postinterventional) microbiome sampling for potential additional analyses in the future.
Author Response
Authors would like to thank you for review of our manuscript and positive opinion. Unfortunately, your valuable suggestions regarding post-interventional visit, and collection of the microbiome sampling for potential analyses in the future cannot be included in the current study protocol because of formal constraints. Current protocol was approved by Ethics Committee, and registered in Clinicaltrials.gov. However, the authors of the manuscript are aware of the importance of the suggestions proposed by the Reviewer, therefore in the section "Expected Results and Discussion” emphasized that the lack of observation after the intervention as well as the lack of analysis of the intestinal microbiome are limitations of the presented study protocol (lines 450-452).